# Real-Time Cutting Temperature Measurement in Turning of AISI 1045 Steel through an Embedded Thermocouple—A Comparative Study with Infrared Thermography

**Bruno Guimarães** [1,2,*], **José Rosas** [3], **Cristina M. Fernandes** [4], **Daniel Figueiredo** [4], **Hernâni Lopes** [3], **Olga C. Paiva** [3], **Filipe S. Silva** [1,2,†] and **Georgina Miranda** [5,†]

1 Center for MicroElectroMechanical Systems (CMEMS-UMinho), University of Minho, Campus de Azurém, 4800-058 Guimarães, Portugal
2 LABBELS—Associate Laboratory, 4710-057 Braga, Portugal
3 CIDEM-DEM/ISEP, School of Engineering, Polytechnic Institute of Porto, Rua Dr. Roberto Frias, 712, 4200-465 Porto, Portugal
4 Palbit S.A., 3854-908 Branca, Portugal
5 CICECO, Aveiro Institute of Materials, Department of Materials and Ceramic Engineering, University of Aveiro, 3810-193 Aveiro, Portugal
* Correspondence: brunopereiraguimaraes@hotmail.com or bguimaraes@dem.uminho.pt
† These authors contributed equally to this work.

**Abstract:** During machining processes, a high temperature is generated in the cutting zone due to deformation of the material and friction of the chip along the surface of the tool. This high temperature has a detrimental effect on the cutting tool, and for this reason, it is of the utmost importance to assess the cutting temperature in real time during these processes. Despite all the advances and investigation in this field, accurately measuring the cutting temperature remains a great challenge. In this sense, this work intends to contribute to solving this problem by experimentally evaluating the potential of the developed approach for embedding thermocouples into the rake face of cutting tools for measuring cutting temperature in real time during dry turning of AISI 1045 steel for different cutting parameters and comparing the obtained results with infrared thermography measurements at the exact same point. A well-defined, smooth micro-groove with good surface quality was produced by laser surface modification. Then a laser-welded K-type thermocouple was fixated in the micro-groove with a MgO ceramic adhesive, ensuring protection from wear and chips, which allowed the creation of WC-Co cutting inserts with the ability to measure cutting tool temperature with a maximum error of 0.96%. Results showed that, despite yielding the same trend, the tool temperature measured by the IR thermographic camera was always lower than the temperature measured by the K-type embedded thermocouple. The proposed embedded thermocouple method proved to be a reliable, precise, accurate, and cost-effective approach for real-time temperature measurement capable of providing useful information for cutting parameter optimization, thus allowing increased productivity and tool life.

**Keywords:** WC-Co cutting insert; cutting temperature; embedded thermocouple; infrared thermography; temperature measurement; turning

## 1. Introduction

Machining processes are not yet completely understood due to the highly non-linear nature of the process and the complex interaction between deformation and temperature [1]. During machining processes, it is estimated that approximately 90% of the mechanical work applied to the workpiece is transformed into thermal energy, generating a very high temperature in a small area of the cutting tool (cutting zone) [2–4]. This high temperature strongly influences tribological phenomena and adhesion, tool wear, tool life, workpiece surface integrity and quality, and chip formation mechanisms, leading to high operating

costs and reduction of the end-product quality [1,5,6]. It is a well-proven fact that cutting parameters, namely depth of cut ($a_p$), feed ($f_n$), and cutting speed ($v_c$) have a distinct effect on heat generation and, consequently, on the lifetime of cutting tools, with the cutting speed having the greatest influence, followed by feed and depth of cut [7,8]. As the tool wear increases, the generation of heat also tends to increase, due to a contact area increase at the tool–chip and tool–workpiece interface because of flank and crater wear. This leads to a reduction of the cutting process efficiency, therefore requiring more energy to perform the cutting action, consequently leading to higher temperatures [1,9,10].

Therefore, it is of the utmost importance to accurately measure cutting temperature in real time to effectively and adequately optimize cutting parameters and cutting fluid flow for minimizing heat generation, temperature, and tool wear [9,11].

Despite all the technological advances and intensive investigation in this field, accurately measuring cutting temperature during machining processes still remains a great challenge, mainly due to workpiece or cutting tool movement, small contact areas involved, chip obstruction on the rake face, and large wear mechanisms [12,13].

Temperature measurement methods can be divided into two main categories: conduction and radiation. For conduction, interaction of energy among the particles from higher energy to lower energy occurs, while radiation methods depend upon the emissivity and temperature of the body surface [14]. These methods include embedded, tool-work, traverse, single-wire and thin-film thermocouple, thermal paints, fine powders, PVD coatings, metallographic methods, infrared (IR) pyrometry, and infrared thermography. Despite all the available methods, none can be considered flawless and able to provide accurate results in all situations, each having its own advantages and disadvantages [2,9,15].

Thermocouple methods are among the most widely used for measuring temperature during machining processes because of their low cost, robust nature, simplicity of installation, and capability to operate over a wide temperature range [6,16]. Temperature measurement by a thermocouple is based on the Seebeck effect, i.e., a small thermoelectric current (emf) is generated when two dissimilar metals, alloys or nonmetals are joined at one end (hot junction), and at the other end (cold junction), the open circuit voltage or Seebeck voltage is measured. This voltage is a function of the temperature difference between the hot and cold junctions, as well as the Seebeck coefficients of the two materials used [17,18]. Among these, the embedded thermocouple method was stated to be one of the most advantageous, since the creation of micro-grooves/holes for thermocouple allocation in the cutting tool allows the achievement of reliable thermocouple protection from flowing chips, as well as a rapid temperature response [19,20].

Chen et al. [21] employed the embedded thermocouple method to measure the peak temperature on machined surfaces in hard turning of hardened steels. A K-type thermocouple was placed in a micro-groove machined cavity, right behind the cutting edge in the flank face, and protected with welded aluminum. Zhao et al. [22] measured the internal temperature variation in orthogonal cutting of Inconel 718 with uncoated and coated WC-Co cutting tools by using an embedded K-type thermocouple placed in a hole drilled by electrical discharge machining. Xie et al. [23] used the embedded thermocouple method to evaluate the influence of micro-groove shape and size on rake face temperature during dry turning of titanium alloy with non-coated cutting tools. Thermocouples were installed in blind holes at 0.6 mm distance from the tool rake face. Campidelli et al. [24] developed a Bluetooth wireless transmission system capable of monitoring the temperature during milling of AISI D2 steel with a tungsten carbide cutting tool by embedding a K-type thermocouple in a fabricated tool hole.

Infrared thermography is another method widely used for cutting temperature measurement, due to its ability to visualize thermal gradients, operate without contact, and fast response. However, the accuracy of this method is highly dependent on the optical properties of the target object, cannot be used when cutting fluid is employed, and chip obstruction makes it difficult to measure the temperature at the tool-chip interface [25,26].

Heigel et al. [27] measured the temperature at the tool–chip interface with an infrared thermographic camera while machining Ti-6Al-4V with a transparent cutting tool. This allowed measurement of the radiant temperature near the cutting edge, as well as observation of the chip curl and breakage. Hao et al. [28] compared the cutting temperature of uncoated and TiAlN coated cutting tools in orthogonal cutting of H13 hardened steel by using an IR thermographic camera. Ramirez–Nunez et al. [29] used infrared thermography to develop a smart sensor for the timely detection of tool rupture in the milling process in dry and with coolant conditions, being the smart sensor capable of providing heat curves, the thermogram, and the thermogram's average acquired during the monitoring of the cutting process. Pratas et al. [30] monitored the temperature during face milling of an Inconel 718 workpiece with a diamond coated cemented carbide end mill by infrared thermography and compared it with a boron doped diamond-based thermistor fabricated by HFCVD.

Additionally, other methods have been employed for rake face cutting temperature measurement. Li et al. [20] reported a novel fabrication process for embedding K-type thin-film thermocouples into the rake face of cemented carbide cutting tools. Kesriklioglu et al. [31] fabricated thin-film thermocouples in a commercially available tungsten carbide cutting insert rake face for accurately measuring the tool–chip interface temperature during turning of AISI 1040 steel. Han et al. [32] obtained the real-time temperature in the cutting edge during turning by using a near-infrared fiber-optic two-color pyrometer inserted into a micro-hole below the tool rake face. Saelzer et al. [33] utilized a small two-color fiber-optic pyrometer with a diameter of 330 μm to directly measure the tool rake face temperature in orthogonal cutting of AISI 1045 steel and Ti-6Al-4V by preparing a workpiece with three slots for locally and temporarily interrupting the chip and, consequently, allowing measurement of the rake face tool temperature. This strategy was then employed by Afrasiabi et al. [34] to validate the results obtained by numerical simulation.

In this study, an embedded K-type thermocouple approach was developed for rake face cutting tool temperature measurement during dry turning of AISI 1045 steel. This approach differs from other embedded thermocouple studies reported in the literature because it doesn't resort to the creation of holes through the cutting tool, that can weaken the overall tool strength, to measure the temperature in the rake face [16,22,24,35,36]. Different cutting parameters were evaluated to assess tool temperature, with the obtained results being compared with infrared thermography measurements at the exact same point, thus providing useful information for cutting parameter optimization and, consequently, allowing increased tool life and productivity.

## 2. Materials and Methods

### 2.1. Concept Design

Cemented carbides, notably WC-Co, are the most widely used materials for cutting tools due to their inherent high hardness, toughness, and wear resistance [37,38]. For this reason, uncoated WC-Co commercial cutting inserts (CNMG 120408-GS, Palbit S.A., Branca, Portugal) were chosen to be used in this study. A K-type thermocouple was selected to be embedded in the rake face as close as possible to the cutting edge, but outside the cutting zone so as to not affect the cutting process. As shown in Figure 1, the thermocouple tip was placed at 2.9 mm from the cutting edge in a micro-groove produced specially for this purpose. High-temperature ceramic adhesive was used for thermocouple protection, insulation, and fixation in the micro-groove.

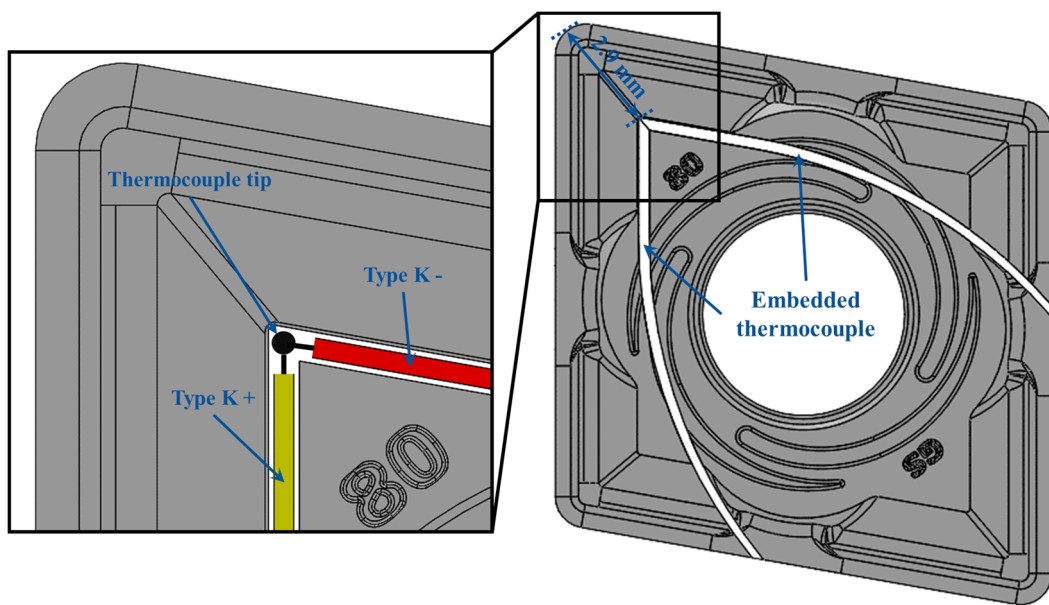

**Figure 1.** Schematic representation of the K-type embedded thermocouple WC-Co insert.

*2.2. Micro-Groove Production*

A Nd:YVO4 fiber laser (XM-30D, XianMing Laser, Liaocheng, China) with a maximum working power of 30 W, a laser spot size of 10 μm, a pulse width of approximately 10 μs, and a wavelength of 1064 nm was used to produce a micro-groove on the rake face of the WC-Co cutting inserts. The micro-groove was defined by using a computer-aided design system, with the laser parameters being optimized to obtain suitable dimensions for further thermocouple embedding and a better surface quality and smoothness. This process led to the selection of the parameters presented in Table 1. During laser processing, a jet of air was used to remove the produced debris and then all samples were ultrasonically cleaned in isopropyl alcohol for 5 min and airdried to remove any remaining debris and contaminants resulting from the process.

**Table 1.** Laser parameters used for micro-groove production.

| Parameter | Value |
|---|---|
| Laser power (W) | 30 |
| Scan speed (mm/s) | 2000 |
| Number of passages | 150 |
| Pulse repetition rate (kHz) | 20 |
| Wobble diameter (mm) | 0.5 |
| Wobble distance (mm) | 0.02 |

*2.3. Embedded Thermocouple Production and Calibration*

K-type thermocouples were manufactured by laser welding (LM-DVO 100, Sisma, Vicenza, Italy) the positive and negative leg wires (0.13 mm diameter), thus creating the hot junction. To ensure the thermocouple fixation in the micro-groove, as well as insulation between the thermocouple and insert material and thermocouple protection from wear and chips, a MgO ceramic adhesive (Resbond 906, Final Advance Materials, Didenheim, France) with a maximum working temperature of 1650 °C was used. The characteristics of MgO, such as high dielectric strength, durability, malleability, quick response to temperature fluctuations, mechanical stability at high temperatures, and extremely high melting point

(2852 °C), make it the preferred option for thermocouple insulation material in a multitude of temperature measurement applications [39,40].

The WC-Co inserts with the laser-welded K-type embedded thermocouple in the rake face were then calibrated by using PREZYS T-25N equipment. For the calibration tests, heating from 25 to 125 °C and then cooling from 125 °C to 25 °C at intervals of 10 °C with a stabilization time of 240 s for each step were performed.

### *2.4. Tool Temperature Measurement*

The WC-Co cutting inserts with a K-type embedded thermocouple were then used to assess the tool temperature during single pass dry turning of AISI 1045 steel for different cutting parameters (see Table 2). These tests were carried out in a lathe machine (Leopard I, Gornati, Legano, Italy) for an evaluation length of 150 mm; the thermocouples were connected to a National Instruments signal acquisition system module (NI 9213), and three tests were performed for each experiment.

**Table 2.** Dry turning tests cutting parameters.

| Experiment | Cutting Speed (m/min) | Feed (mm/rev) | Depth of Cut (mm) |
|:----------:|:---------------------:|:-------------:|:-----------------:|
| 1 | 40 | | 1 |
| 2 | 70 | | 1 |
| 3A | 130 | 0.246 | 1 |
| 3B | 130 | | 2 |
| 4 | 160 | | 1 |

Simultaneously, the temperature was monitored by an IR thermographic camera (FLIR E60, Teledyne FLIR, Wilsonville, OR, USA) that was placed above the tool rake face to measure the same exact point as the embedded thermocouple. The emissivity of an object is a crucial parameter when determining its temperature by IR thermography, the emissivity being dependent on the composition of the material and its temperature. Based on the literature [41–43], and since, in this study, uncoated WC-Co cutting inserts were used, thus presenting a matt surface, a value of 0.90 was defined for the emissivity. The temperature registered by the IR thermographic camera was then correlated with the temperature measured by the embedded thermocouple over the machining time.

### 3. Results and Discussion

In the present study, a micro-groove was produced on the rake face of uncoated WC-Co cutting inserts by laser surface modification in order to embed a K-type thermocouple for tool temperature measurement during dry turning of AISI 1045 steel. As shown in Figure 2a,b, the produced micro-groove matched the defined design (see Figure 1), being well-defined, smooth, and with good surface quality. Additionally, no microcracks, spatter, and heat-affected zones were observed, allowing to conclude that the laser parameters for micro-groove production were properly selected so as to not affect the cutting insert properties. The micro-groove surface exhibited some scanning marks, due to the action of the laser, which is potentially beneficial for the embedding process. As reported in the literature, surface roughening plays an important role in the bonding strength, since a larger bonding area contributes to a strength increase of the adhesive joint [44,45].

Figure 2c depicts the micro-groove surface topography analyzed by 3D optical profilometry, and the micro-groove profile is shown in Figure 2d. The laser parameters, presented in Table 1, were chosen to obtain a groove capable of guaranteeing a successful embedding of the K-type thermocouple, with an achieved micro-groove width and depth of 468.0 ± 7.7 μm and 439.3 ± 19.9 μm, respectively.

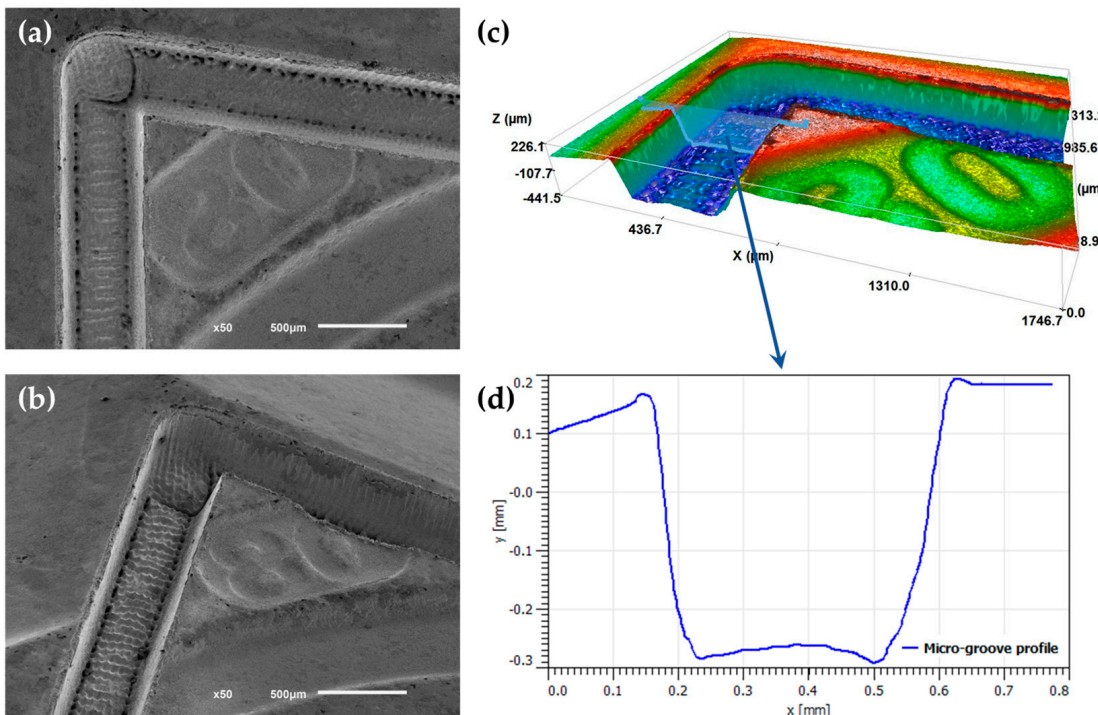

**Figure 2.** Micro-groove produced by laser surface modification: (**a**) top view SEM image; (**b**) tilt 45° SEM image; (**c**) 3D optical profilometry; and (**d**) surface topography profile.

Laser welding proved to be a viable method for thermocouple production, since it was able to form a strong hot junction with a spherical shape, as shown in Figure 3a. This type of junction, exposed junction, allied with the high thermal conductivity of the MgO ceramic adhesive, allows greater sensitivity and quicker response time when compared to an insulated or grounded junction [46,47]. A hot junction diameter of approximately 250 μm was obtained, which is in line with the desirable dimensions to be fitted in the produced micro-groove. In Figure 3b it is possible to observe the K-type embedded thermocouple in the WC-Co cutting insert with the high temperature MgO ceramic adhesive.

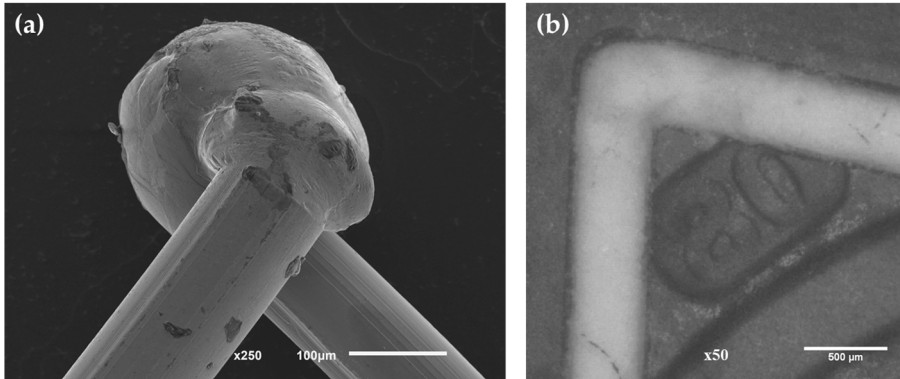

**Figure 3.** (**a**) SEM image of the laser-welded K-type thermocouple hot junction; (**b**) SEM image of the WC-Co cutting insert with the K-type embedded thermocouple.

Figure 4 shows the temperature measured by the K-type embedded thermocouple in comparison with the reference thermocouple during the calibration test. To evaluate the reproducibility, three tests were performed in different WC-Co cutting inserts, showing the measurements' repeatable results. For the evaluated temperature range, a maximum error of ±1.2 °C was found, representing an error of 0.96% of the maximum temperature measured by the system (125 °C), which attests to the reliability and accuracy of the laser

welding and embedding processes for producing WC-Co cutting tools capable of measuring cutting temperature in real time.

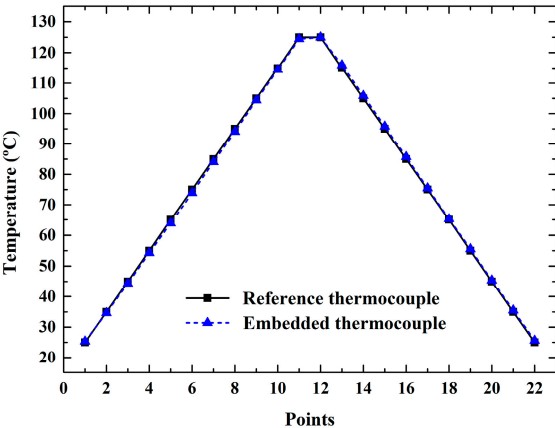

**Figure 4.** Calibration test of the K-type embedded thermocouple.

The influence of different cutting parameters, namely cutting speed and depth of cut, on the tool temperature during turning of AISI 1045 steel was measured by the developed K-type embedded thermocouple and the obtained results were compared with infrared thermography measurements at the exact same point.

Figure 5 depicts the effect of cutting speed variation on the tool temperature for both methods at a constant depth of cut of 1 mm. As expected, the higher the cutting speed, the higher the tool temperature generation, with both methods producing this outcome. However, despite yielding the same trend, as depicted by curve fitting, the tool temperature measured by the IR thermographic camera was always lower than the temperature measured by the K-type embedded thermocouple. This is due to some difficulties encountered during the measurement process, such as chip obstruction of the measuring point and/or a defined value of emissivity that was lower than the real one, which leads to a lower temperature reading by the IR thermographic camera [16]. Additionally, there is a tool temperature increase with the machining time, but a tendency to reach an equilibrium temperature is observed if the cutting time is long enough.

At the end of the evaluation length, cutting is halted, leading to a sudden decrease in temperature, with the maximum tool temperatures measured by the K-type embedded thermocouple being 178.07 °C, 246.65 °C, 256.36 °C, and 327.25 °C, for cutting speeds of 40 m/min, 70 m/min, 130 m/min, and 160 m/min, respectively. The similar maximum tool temperature obtained for cutting speeds of 70 m/min and 130 m/min, was due to the evaluation length not being enough to reach the temperature equilibrium, since a higher cutting speed leads to a lower machining time, meaning that higher cutting speeds need more time to reach the maximum tool temperature. As shown in Figure 5, the maximum tool temperature was reached for experiment 2 at 86.2 s of machining time, while for experiment 3A, the same temperature occurred at 40.0 s of machining time. It is also worth mentioning that for experiment 4, only 15.0 s were necessary to reach this temperature, while for experiment 1, the evaluation length was not sufficient.

In Figure 6, the effect of depth of cut variation on the tool temperature, for both methods, is shown for a constant cutting speed of 130 m/min. It is possible to observe that the tool temperature increased with the depth of cut increase, since a higher cutting depth leads to more material being deformed, leading to a higher heat generation in the cutting zone. Maximum tool temperatures of 256.36 °C and 340.03 °C were measured by the K-type embedded thermocouple for a depth of cut of 1 mm and 2 mm, respectively.

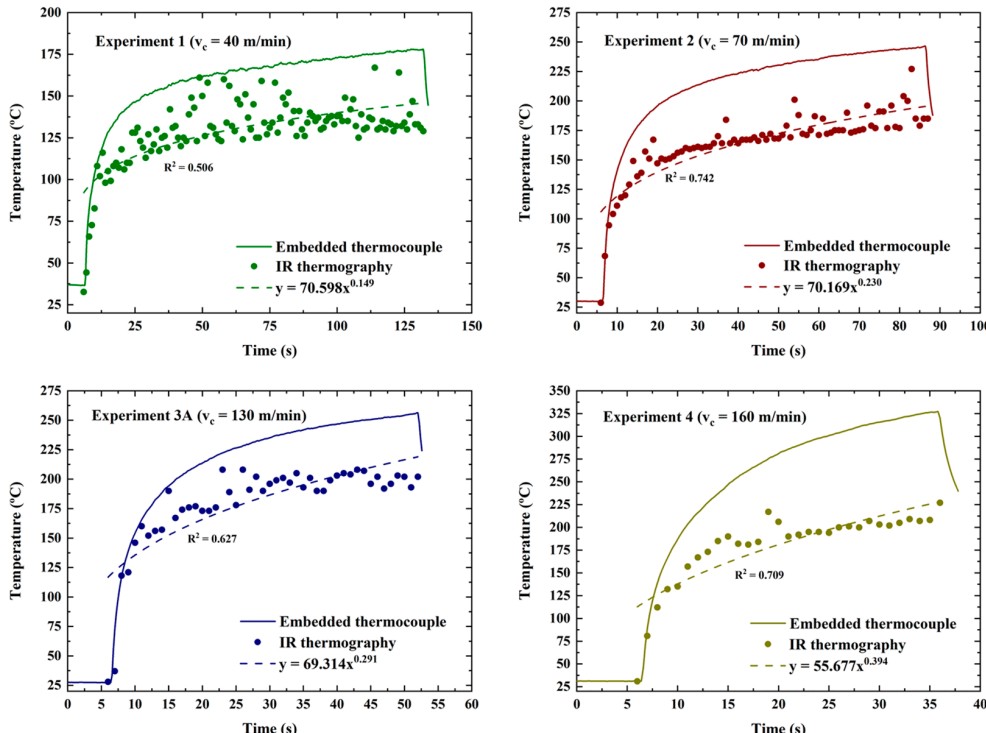

**Figure 5.** Effect of cutting speed variation on the tool temperature over the machining time ($a_p$ = 1 mm).

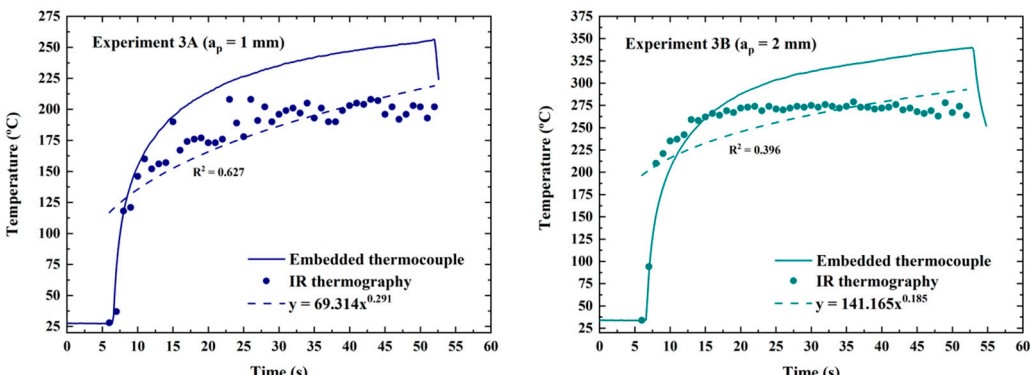

**Figure 6.** Effect of depth of cut variation on the tool temperature over the machining time ($v_c$ = 130 m/min).

The IR thermographs obtained over some machining times and the embedded thermocouple measurements show that the cutting tool heated up very quickly in the initial seconds, but over the machining time, this rate tended to decrease (see Figure 7). Additionally, this rate of temperature rise increased as the cutting speed and cutting depth increased, as shown in Figures 5 and 6. In Figure 7, it is also possible to observe the temperature distribution over the tool's surface, the chip formation, the K-type embedded thermocouple, the workpiece, and the cutting zone.

By comparing both measurement methods, it is possible to conclude that the obtained results highlight some of the disadvantages of the infrared thermography method, since smooth, accurate, and reproducible measurements were difficult to obtain. On the other hand, the proposed embedded thermocouple method seems to be a reliable, precise, accurate, and cost-effective approach for real-time temperature measurement. These results also allow us to conclude that the MgO ceramic adhesive ensured efficient thermocouple fixation as well as insulation and protection from wear and chips.

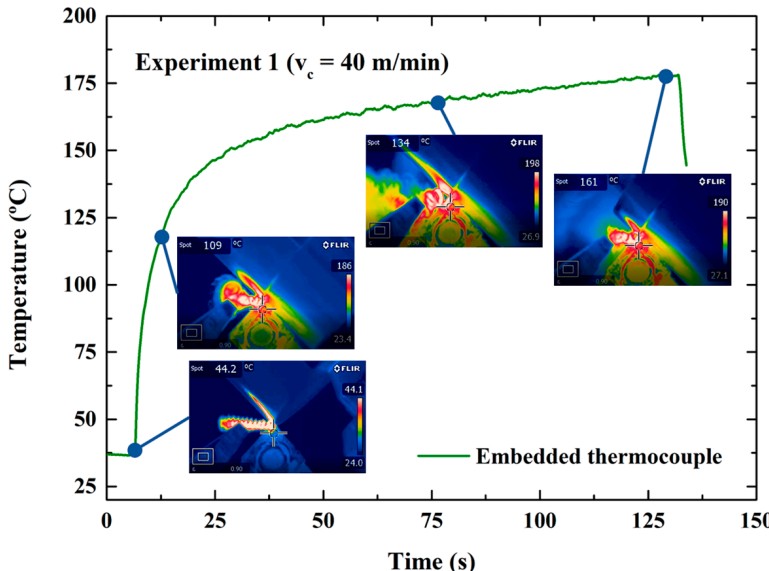

**Figure 7.** Example of IR thermographs obtained for experiment 1 over some machining times.

### 4. Conclusions

- Laser surface modification was able to produce a well-defined, smooth micro-groove with good surface quality matching the defined design for subsequent thermocouple embedding.
- The laser parameters used in this work were properly selected to not affect the cutting insert properties, since no microcracks, spatter, or heat-affected zones were observed.
- Laser welding proved to be a viable approach for producing reliable, accurate, and precise K-type thermocouples with a maximum error of 0.96% of the evaluated temperature and achieving a hot junction diameter of approximately 250 μm.
- WC-Co cutting inserts with the ability to measure cutting tool temperature in real time, with great sensitivity, quick response time, as well as protection from wear and chips, were obtained.
- Despite yielding the same trend, the tool temperature measured by the IR thermographic camera was always lower than the temperature measured by the K-type embedded thermocouple.
- Maximum tool temperatures of 178.07 °C, 246.65 °C, 256.36 °C, and 327.25 °C were measured by the K-type embedded thermocouple for cutting speeds of 40 m/min, 70 m/min, 130 m/min, and 160 m/min, respectively.
- Regarding the depth of cut, maximum tool temperatures of 256.36 °C and 340.03 °C were measured by the K-type embedded thermocouple for a depth of cut of 1 mm and 2 mm, respectively.
- The IR thermographs obtained over some machining times and the embedded thermocouple measurements showed that the cutting tool heated up very quickly in the initial seconds of turning, but over the machining time, this rate tended to decrease.
- The proposed embedded thermocouple method was shown to be a reliable, precise, accurate, and cost-effective approach for real-time temperature measurement, thus providing useful information for cutting parameter optimization, and allowing increased productivity and tool life.
- Additional studies will be performed to develop a reliable approximation of the temperature in the cutting edge by correlating the obtained results in this study with numerical simulations and/or analytical modeling. Additionally, the influence of tool wear, tool coatings, and tool geometry on the tool temperature measured by the approach developed in this study are other research lines that can contribute to the knowledge in this field.

**Author Contributions:** Conceptualization, B.G., D.F. and G.M.; methodology, B.G., J.R. and C.M.F.; validation, B.G. and J.R.; investigation, B.G. and G.M.; resources, D.F., H.L., O.C.P. and F.S.S.; writing—original draft preparation, B.G.; writing—review and editing, J.R., C.M.F., H.L. and G.M.; visualization, B.G.; supervision, C.M.F., F.S.S. and G.M. All authors have read and agreed to the published version of the manuscript.

**Funding:** This work was supported by FCT (Fundação para a Ciência e a Tecnologia) through the grant 2020.07155.BD and by the project POCI-01-0145-FEDER-030353 (SMARTCUT). Additionally, this work was supported by FCT national funds, under the national support to R&D units grant, through the reference projects UIDB/04436/2020 and UIDP/04436/2020. Finally, this work was also developed within the scope of the project CICECO-Aveiro Institute of Materials, UIDB/50011/2020, UIDP/50011/2020 & LA/P/0006/2020, financed by national funds through the FCT/MEC (PIDDAC).

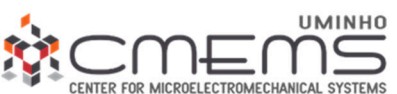 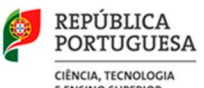 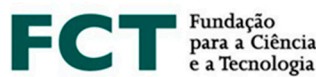

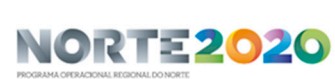 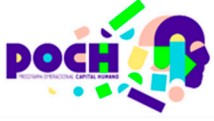 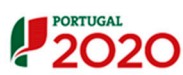 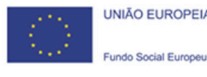

**Data Availability Statement:** The data presented in this study are available on request from the corresponding author.

**Conflicts of Interest:** The authors declare no conflict of interest.

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
