# Peer review of "Real-Time Cutting Temperature Measurement in Turning of AISI 1045 Steel through an Embedded Thermocouple—A Comparative Study with Infrared Thermography"

_jmmp, doi:10.3390/jmmp7010050_

Round 1

Reviewer 1 Report (New Reviewer)

Cutting processes already enjoy an extensive know-how in the manufacturing domain, having been intensively studied in the last century by researchers and also by industrial companies. Even so, there are many research topics in this field that need and deserves future investigations in order to be improved or to solve different issues. This paper presents a research regarding the real time measuring of cutting temperature during a dry turning process of AISI 1045 steel.

First chapter: This topic is a research theme approached by various researchers in cutting process field, this being properly summarized in the introduction chapter of this paper (temperature measuring methods and technique adopted in the literature).

Second chapter: Even so, in this paper, the authors presents how a K-type thermocouple is embedded in a micro-groove near the cutting edge and clamped using high temperature ceramic adhesive. The improvement bring by this method is that no more holes should be drilled in the cutting insert which can weaken its stiffness during cutting process. The micro-groove were obtained by laser fibre processing (well described in this chapter). It is also described how the K-type thermocouple was welded in the micro-groove and also how the calibration process was done. The process setup and parameters are presented and also the measurement method. An additional temperature method was used above the tool rake face (inflared thermographic camera) beside the thermocouple, in order to correlate the temperatures.

Third chapter: This chapter is a simple one but contain all the essentials needed to describe the measurements and the observations upon the experiment. The micro-groove processed in the cutting insert is analysed, the thermocouple laser welding process is well commented. Both type of results (measurements made by IR thermographic camera and the measurements made by the K-type embedded thermocouple) were presented in many comparison and the important facts are well highlighted. A claim is made as a great advantage for the K-type embedded thermocouple method, among others, that the MgO ceramic adhesive ensured a well insulation and protection from wear and chips. For sure, the infrared thermography method has also his well-deserved benefits.

Fourth chapter: This chapter presents the basic conclusions regarding the research but I think it should be improved (check the recommendations for authors made by reviewer). The conclusion chapter is a bit too superficial and should be improved.

Overall, the paper is quite well structured in 4 chapters and several subsections. Even so, its content can be improved, especially for the conclusion chapter which briefly presents only several important aspects presented in the paper, but not as it should be in a conclusion chapter.

Thus, below are several recommendations for authors:

- maybe in the introduction it should be good to present also another founding claimed by other research studies (I meen the influence factors upon the temperature on a turning process)

- maybe should mention if the laser fibre heating affect the cutting insert properties (If NO, why no ? If YES, which are the influences). This can be mentioned when is presented the micro-groove processing by laser fiber;

- the first paragraph from “conclusions” chapter tend to present an short overview upon the entire paper. This practice should be avoided because in this way will repeat something that was already presented in the paper content.

- the second paragraph from “conclusions” chapter presents general conclusions regarding the cutting process parameters which influences the increase in temperature. I my opinion, those general conclusions are already well-known and are claims from many others research studies made many years ago. The suggestion is to move those general parameter influences on the introduction chapter (by citing some literature references), and in the conclusion chapter it would be better to insert some particularly values (measurements) obtained on this research, for the presented experiments.

- because the paper concentrate upon the measuring method and no on the factors influence upon the temperature, another remark is that the conclusion should contain more accurate/exact/clear findings/ novelties (claims) upon the measurement method (even using bullets) using K-type embedded thermocouple.

- is recommended to present future research work at the end of the paper.

Author Response

Reviewer 2 Report (New Reviewer)

Please address the following comments before re-considering your manuscript for publication in JMMP.

Abstract

1. The abstract contains some general information about the problem at hand (e.g., % of the mechanical energy dissipation and necessity of precise temperature measurements), which can be omitted or reduced. Please revise and provide more specific details about the key findings of your study, limitations, etc.

2. The English and readability must be improved as well.

1. Introduction

1. The authors seem to have overlooked some of the most relevant studies on the very same topic. In-process temperature measurement in metal cutting is a well-known problem, and various experimental methods have already been developed for resolving it. Please enhance the literature survey by, for instance, referring to the experimental section of this article:

https://doi.org/10.3390/met11111683

and by covering some of its relevant references which deal with a similar challenge in metal cutting temperature measurements. The necessary citations should be included in the introduction section.

2. The purpose of introduction is not just to mention “what” approach has been used by the authors but, more importantly, to mention “why”. Please clarify the rationale behind the choice of your methodology (compared to what already exists in the literature!) and explain in the main text which issues from the existing literature will be addressed by your proposed solution.

2. Materials and Methods

1. Please add a schematic diagram of turning, to which the K-type thermocouple (as in Fig 1) can be directly lined.  

3. Results and Discussion

1. In Fig. 2d: you need to provide more details about the roughness calculation.

2. Please also mention at which plane/section the roughness plot is being presented.  

3. In order for the reader to realize the effects of cutting process geometry (speed & depth) on the tool temperature, please add two separate plots (or bar-charts), where the tool temperature would be a function of steady-state speed and cutting depth values – instead of horizontal time axis.  

4. Conclusions

1. The “Conclusion” section must be improved. Please cast the key findings of your research into a bullet-point format and point out the limitations/shortcomings of your approach as possible directions for future development. It is important to suggest a few opportunities to further develop the approach proposed in your study.

2. Clearly, one of the main shortcomings of the present study is its foundation on a purely experimental basis without incorporating any modeling/simulation insights into the framework. This needs to be stated clearly in the conclusion.

General Remarks

The English language of this paper in its current form is OK in most places. Nonetheless, there are still some punctuation issues, missing articles, wrong structures, and grammatical errors. Please have a native English speaker proofread your text or use a professional language editor.

Author Response

Reviewer 3 Report (New Reviewer)

1. Most of the references are relatively old, pleaes update it.

2. How to calibrate/validate the temperature measured with infrared thermography? 

3. Obviously temperatures measured by IR are lower compared with corresponding results from embedded thermocouple, the scientifc contribution of this work was not presented in details.

4. Did the authors consider the effect of tool wear on temperature? All new inserts were used for each trial?

5. Please shorten the conclusions with key findings/summary from the experimental works.

6. Why was the data recorded from embeded thermocouple presenting as continous line, while the ones from IR were individual points in Figs. 5 and 6?

Author Response

Reviewer 4 Report (New Reviewer)

Figure 4 shows the course of temperatures during calibration, measured by the reference thermocouple and the K-type thermocouple - a rare 100% match. That's all right?

In the end, you state how the evaluation, that it was confirmed, that when the cutting speed or depth of cut is increased, the temperature increases. However, this statement is a well-known fact, which you also mentioned in the literature search. But you do not state by how much and what this implies for the given experimental verification, e.g. thermal effect on the machined surface and what...

Readers would certainly be interested in why there is a difference in the measured temperatures in the cutting zone of the thermocouple and the thermal camera.

Author Response

This manuscript is a resubmission of an earlier submission. The following is a list of the peer review reports and author responses from that submission.

Round 1

Reviewer 1 Report

Notes in the attachment.

Reviewer 2 Report

p. 2, l. 11: „leading to thermal deformation of cutting tool” -> the thermal deformation of the tool usually dos not play a role in cutting processes

The state-of-the-art should be covered more carefully. There are papers available, dealing with direct measurements at the rake face. The technique using two colour fibre pyrometers is also not mentioned.

The paper focusses on the technical issues to integrate the thermocouple in the tool by laser grooving and bonding. The measured results are all well known and could be expected. Using the proposed method, the measurement takes place far away from the cutting zone, as it is the case in applying thermocouples into holes in the tool. A comparison to this often used method is not provided by the authors. So no significant improvement with respect to known methods can be found. It is also known, that measurements directly at the tool is more accurate than measurements by a camera system far away. Therefore, the novelty of the presented scientific results is very limited, if present at all, so I suggest to reject the paper.

Reviewer 3 Report

The reviewer comments of the paper «Real-time cutting temperature measurement in turning of AISI 1045 steel through an embedded thermocouple and infrared thermography: a comparative study» - Reviewer

The authors presented an article «Real-time cutting temperature measurement in turning of AISI 1045 steel through an embedded thermocouple and infrared thermography: a comparative study». The article may have the potential for publication, but in its current form, the novelty is insufficient. A simple comparison of two temperature determination methods is hardly sufficient for publication in reputable international journals. In its current form, this is a good rationale for a new patent. And it may be interesting to beat for practical significance. But now it is primitive for a scientific article. There are several points in the article that require further explanation.

The topic sounds strange as a comparison of two methods of measuring temperature. Now, if the correlation of the increase in temperature with the value of flank wear would be shown, then we could talk about real scientific significance. Of course, in this case, SEM of flank wear is needed. In addition, a clear justification of the physics of cutting is needed. How cutting conditions affect temperature and flank wear. What is the temperature measurement error for different methods and why do they differ? What is the material of the cutting part and what are the physical and mechanical properties, including temperature? The same thing is important to know for harvesting. Who owns the embedded thermocouple? If the authors, then this is one thing. If other researchers, then what is new in this practical approach?

Thus, in its current form, the article has no scientific and practical value and should be rejected.